# Impact of Family Separation on Subjective Time Pressure and Mental Health in Refugees from the Middle East and Africa Resettled in North Rhine-Westphalia, Germany: A Cross-Sectional Study

**DOI:** 10.3390/ijerph182111722

**Published:** 2021-11-08

**Authors:** Matthias Hans Belau, Heiko Becher, Alexander Kraemer

**Affiliations:** 1School of Public Health, Bielefeld University, 33501 Bielefeld, Germany; alexander.kraemer@uni-bielefeld.de; 2Institute of Medical Biometry and Epidemiology, University Medical Centre Hamburg-Eppendorf, 20246 Hamburg, Germany; h.becher@uke.de

**Keywords:** family separation, time pressure, mental health, health disparities, refugees

## Abstract

Little is known about social determinants among refugees resettled in Germany. This study aims to examine the impact of family separation on refugees’ subjective time pressure and mental health. Data come from the FlueGe Health Study (*n* = 208), a cross-sectional study administered by Bielefeld University. We used logistic regression analysis to investigate the effect of family separation on (i) being time-stressed and (ii) having a high risk for adverse mental health, considering sociodemographic and postmigration factors. As a result, more than 30% of participants with a spouse or partner and about 18% with a child or children reported separation. Multiple logistic regression showed that family separation was not associated with being time-stressed, but separation from at least one child was associated with adverse mental health (OR = 3.53, 95% CI = [1.23, 10.11]). In conclusion, family separation primarily contributes to adverse mental health among refugees from the Middle East and Africa resettled in North Rhine-Westphalia, Germany. Therefore, policies and practices that facilitate family reunification can contribute significantly to the promotion of refugees’ mental health and well-being.

## 1. Introduction

In 2015, the ongoing wars and conflicts in the Middle East and Africa led to a dramatic increase in refugees making their way to Europe crossing the Mediterranean [1]. Most of the refugees originated from Syria, Afghanistan, Iraq, and Central Africa. Many of them have been subjected to stressful and adverse experiences on the individual, family, and community level, as reflected in high rates of mental health issues during the postsettlement period [2,3,4,5]. There is a large body of literature showing that refugees experience mental rather than physical impairments [5,6,7]. A study published by the German Federal Chamber of Psychotherapists showed that about half of adult refugees residing in Germany suffered from mental illnesses such as post-traumatic stress disorder (PTSD) and depression [2]. In addition, the literature reveals that prevalence rates in mental disorders were frequently increased in war refugees, even many years after resettlement [8]. Therefore, it seems important to understand the factors that predict postmigration stress and adverse mental health in order to promote refugees’ long-term mental health.

Many refugees are dealing with loneliness and the experience of loss [9,10]. Refugee families are often separated by conflict-induced displacement [11,12] or the migration policies of the host country [13]. In some cases, family members are left behind to seek asylum in the hope of eventual reunification [12]. In Germany, family reunification is linked to a residence or settlement permit [14]. Due to the growing number of asylum applications in 2015 and 2016 [1], the duration for the decision-making in asylum procedures increased from 8 months on average in 2015 [15] to 18 months on average in 2018 [16], which in turn led to delays in family reunification applications. This situation can be seen as a serious threat leading to high levels of psychological stress, as those affected have limited coping capacities in the host country, e.g., family support resources, which is in line with Lazarus’ psychological stress and coping theory [17]. Consistent with Hobfoll’s conservation of resources theory [18], which focuses on the change and conservation of resources in the context of environment and social processes, we argue that refugees separated from their family experience time pressure as a type of psychological stress. Time pressure occurs when a person has less time available (real or perceived) than is necessary to complete a task or obtain a result [19], such as family reunification. Evidence suggests that the presence of a family member in an individual’s postmigration country has a positive effect on postmigration stress [20]. In contrast, family separation is shown to be associated with reduced health-related quality of life (HRQoL) [21,22]. Furthermore, a lack of information about family members left behind is associated with mental illnesses such as depression, somatization, anxiety, and PTSD, while information about family members left behind is associated with better self-rated health [3]. Thus, we hypothesized that refugees who are separated from their family members (particularly from spouse or partner and/or child or children) experience adverse mental health and time pressure as a type of psychological stress after resettlement.

Time pressure has one objective dimension and one subjective dimension [23]. The objective dimension embraces a measurable time shortage, e.g., not having time for an activity, while the subjective dimension is a predominantly subjective emotional experience of fragmented time, demands to do things faster, or feeling rushed. To the best of our knowledge, no study so far aimed to explore the association between family separation and subjectively perceived time pressure, a potential social determinant of health [24], in refugee populations. Previous research on subjective time pressure showed associations with mental health problems such as anxiety and depression [25,26], and a causal relationship in either direction is conceivable [19,27].

Several studies show that family separation may affect refugees’ mental health [22,28,29] and well-being [30]. A previous study conducted amongst adult refugees resettled in Australia [31] focused on the impact of family separation and worry about family and friends on post-traumatic stress symptoms and psychological distress. The study also examined the contributions of demographic and postmigration stressors, with older age and female sex found to be more consistent predictors than family separation and worry about family and friends. An older study has also shown that a long asylum procedure and a longer stay in the host country can have a negative impact on refugees’ overall health situation and well-being [32]. Concurrently, there are often barriers to receiving medical services and accessing the social system [33]. Although a study by Wetzke et al. [34] shows that primary care is most needed directly after arrival, refugees in Germany still have limited access to medical care during their asylum process [35]. From a public health perspective, information is needed on factors that promote and impair health among refugees seeking protection in Germany.

Thus, the goal of our study was to investigate the impact of separation from spouse or partner and/or child or children as a nuclear family on mental health and subjective time pressure, as it may be a stressor involved in the stress–distress relationship, among refugees in North Rhine-Westphalia, Germany, considering sociodemographic and postmigration factors as the main relevant confounders.

## 2. Materials and Methods

### 2.1. Sample Description and Procedure

Data come from the FlueGe Health Study (FHS), a cross-sectional study administered by Bielefeld University, conducted on refugees from the main countries of origin that contributed to the European refugee crisis in 2015 and 2016 in the region of East Westphalia-Lippe in North Rhine-Westphalia, Germany. The data were collected between February and November 2018 and included personal interviews and physical examinations, carried out by trained interviewers. Informed consent forms, information letters, and the questionnaire were translated into the following five languages by Kantar Public, a consulting and market research institute: Arabic, Farsi, Kurmanji, English, and German. Participants were recruited from shared and private accommodation in eight different locations in East Westphalia-Lippe, with municipal cooperation partners and social workers providing access to potential participants. The FHS included all participants who were willing to participate (convenience sampling) and signed informed consent, excluding those who were younger than 18 years of age; could not speak Arabic, Kurmanji, Farsi, English, or German; or had been in Germany for more than five years. Prior to the interviews, all potential participants were personally informed by the field team about the aims and procedure of the study during an on-site visit with an invitation to participate. The field team consisted of an academic researcher and trained interviewers in the required languages. All potential participants who could not be contacted in person received a letter informing them of the study aims and procedure and asking them to contact the field team by telephone. Approval from the Ethics Commission of Bielefeld University was obtained before the data were collected to ensure ethical and data protection guidelines were followed.

A total of 827 men and women aged 18 to 75 years were assessed for eligibility and invited to the study. Of these, *n* = 130 had an inadequate language level, and *n* = 371 refused to participate in the study. The main reasons were personal reasons and no interest in the research. A total of 326 men and women signed informed consent (recruitment rate, 46.8%) and completed the study. Prior to data analysis, the FHS study population was reduced to individuals with a spouse or partner and/or a child or children (*n* = 208).

### 2.2. Measures

#### 2.2.1. Time Pressure

Subjectively perceived time pressure was assessed using a single-item question from the German Socioeconomic Panel (SOEP): “Please think about the last four weeks. How often did it occur within this period of time that you felt rushed or pressed for time?” with five possible responses: “Always”, “Often”, “Sometimes”, “Seldom”, and “Never” [36]. In order to identify associated factors, time pressure was dichotomized with participants who rated “always” or “often” being categorized into the time-stressed group, whereas those who rated “sometimes”, “seldom”, or “never” were categorized into the not time-stressed group, which is consistent with the literature [37].

#### 2.2.2. Mental Health

Mental health was measured using the mental component summary (MCS) score of the Short Form-12 Health Survey-SOEP (SF-12-SOEP) [38]. To compare to published means, the MCS scale was transformed into a range from 0 (minimum) to 100 (maximum), with a higher value indicating a better state of mental health. Additionally, norm-based scoring was performed by first z-transforming MCS scale using factor loadings for weighting served by the SOEP2004 data as the norm population [38] and then transforming them to a mean value of 50 and a standard deviation (SD) of 10. Further, the MCS scale was dichotomized with the sample mean value as a cut-off point to classify people at lower (≥MCS cut-off point of 44.5) and higher risk (<MCS cut-off point of 44.5) for mental health problems. This is consistent with the literature, in which optimal cut-off values to screen for depressive disorders in a general population vary between MCS scores of 42.0 [39] and 45.6 [40].

#### 2.2.3. Family Separation

Family separation was indicated by asking participants the following question: “Where do your (i) spouse or partner and (ii) child or children live now if present?” with predefined response categories: “Here in the facility/flat”, “Nearby, in another facility/flat”, ”Elsewhere in Germany”, “In our native country”, “Elsewhere abroad”, “I don’t know” and “Deceased”. Responses were categorized into separated (elsewhere in Germany, in our native country, elsewhere abroad, and I don’t know) versus not separated (here in the facility/flat, nearby, in another facility/flat).

#### 2.2.4. Sociodemographic and Postmigration Data

Sociodemographic and postmigration information included age, sex, country of origin, residence status according to the German residency law, and length of stay since the arrival to Germany. Information on country of origin was categorized into six groups: Syria, Afghanistan, Iraq, Iran, African countries, and other countries. Residence status was categorized as secure (entitlement to asylum, refugee protection, subsidiary protection, and a national ban on deportation) and insecure (in procedure, temporary suspension of deportation, and a requirement to leave). The length of stay since arrival in Germany was determined using an official proof of arrival and the time of the interview, categorized as <12, 12–24, 25–36, and >36 months.

### 2.3. Statistical Analyses

Statistical analyses were performed using STATA MP in version 16. Descriptive statistics were used to examine participants’ time pressure, adverse mental health, family separation, sociodemographic, and postmigration characteristics. Differences in time pressure and adverse mental health by family separation, sociodemographic, and postmigration factors were analyzed using chi-squared, Fisher’s exact, and *t*-test. Multiple logistic regression [41] was applied to examine whether family separation, sociodemographic, and postmigration factors were potential risk factors for (i) being time-stressed and (ii) having a higher risk for adverse mental health. The null hypothesis was that family separation was not a potential risk factor for (i) being time-stressed and (ii) having a higher risk for adverse mental health considering the main relevant confounders of sociodemographic and postmigration factors. Cases presenting a missing value for at least one of the modeling variables were excluded from analyses (listwise deletion). Nevertheless, we could not find any specific characteristics for incomplete answers. Independent variables considered in the multivariable analyses were age, sex (as a dummy variable (DV)), country of origin (DV), residence status (DV), length of stay since arrival (DV), separation from spouse or partner (DV), and separation from child or children (DV). Hosmer–Lemeshow goodness of fit test was used to evaluate the logistic regression models. We computed odds ratios (ORs) and 95% confidence intervals (CIs), and the significance was set at *p* < 0.05.

## 3. Results

Table 1 presents the characteristics of the study population. The majority were male, most immigrated from war-affected Middle Eastern countries, and the mean (SD) length of time since arrival in Germany was 29.9 (11.6) months. More than 30% of participants with a spouse or partner, and about 18% with a child or children reported separation. About 38.9% of participants subjectively experienced time pressure always (19.4%) or often (19.9%), with a mean (SD) MCS score of 39.6 (13.9). Participants who subjectively experienced time pressure sometimes (28.2%), seldom (8.7%), and never (23.8%) had a mean (SD) MCS score of 47.5 (14.3). Overall mean (SD) MCS score was 44.5 (14.6).

Table 2 provides detailed information on sociodemographic and postmigration factors stratified by time pressure and adverse mental health groups. More than half of the female participants were classified in the higher risk for adverse mental health group, while most male participants reported less time pressure and better mental health. Frequency in the time-stressed group increased with length of stay since arrival in Germany from 29.2% (<12 months) to 44.3% (>36 months). Nevertheless, the majority of participants who reported family separation were in the not time-stressed group but in the higher risk for adverse mental health group.

Concerning family separation, it was found that the majority of respondents with a partner and at least one child were not separated (76.5%), while 11.8% of respondents who were separated from their spouse or partner were also separated from their child or children.

Table 3 shows the regression model between being time-stressed as the dependent variable and independent variables (*n* = 129), which had an acceptable fit (Hosmer and Lemeshow statistic: χ^2^ = 1.87, df = 8, *p* = 0.985). Participants with insecure residence status were more likely to be time-stressed as compared to those with a secure status.

Table 4 displays the final model with factors associated with higher risk for adverse mental health (*n* = 124), also with an acceptable fit (Hosmer and Lemeshow statistic: χ^2^ = 8.63, df = 8, *p* = 0.375). Separation from at least one child was strongly associated with a higher risk for adverse mental health. Furthermore, female sex and insecure residence status were also found to be positively associated with a higher risk for adverse mental health, while a length of stay of more than three years since arrival was negatively associated.

## 4. Discussion

This study aimed to investigate the association of family separation with time pressure and adverse mental health considering the main relevant confounders of sociodemographic and postmigration factors among refugees in North Rhine-Westphalia, Germany. This study showed that separation from at least one child was positively associated with a higher risk for adverse mental health; however, separation from spouse or partner and/or child or children was not associated with being time-stressed.

In our sample, about 38.9% of refugees were classified as being time-stressed; however, one study examining the relationship between socioeconomic characteristics and time pressure in the German general population based on SOEP data in its 2002 wave version using the same instrument [37] showed a lower overall prevalence of being time-stressed (35.3%). Therefore, time pressure is prevalent in our studied refugee sample. As this study may be the first to explore the association between family separation and subjectively perceived time pressure in a resettled refugee population, we did not find support for our hypothesis that refugees separated from their family members experience time pressure always or often. Nevertheless, it is interesting to note that a study primarily focusing on the mental health consequences of family separation for refugees found that separation from family members was a major stressor because family reunification was one of the refugees’ primary needs [22]. This stressor could be exacerbated in times of the COVID-19 pandemic, as asylum and resettlement processes are disrupted by lockdowns [42].

Concerning our second hypothesis that refugees who are separated from their family members experience adverse mental health, we did find support that separation from at least one child might be a source of health inequalities among resettled refugees. Our findings show that separation from a child may be a risk factor for mental illness, which is consistent with previous research highlighting the negative impact of child separation on migration-related stress [43], distress [22], and mental health and well-being [44]. In the contrast, prior research shows that children who were separated from their parents report greater symptoms of anxiety [45], depression [46], and psychotic disorders [47]. The findings of our study extend previous work by demonstrating the adverse impact of child separation in resettled refugees from diverse language and ethnic groups. We also observed that Afghans and Iranians in particular were at higher risk for mental health problems. However, origin itself was not a significant influencing factor in the multivariable analyses.

Concerning sociodemographic and postmigration factors, multiple logistic regression revealed that insecure residence status was positively associated with both being time-stressed and having a higher risk for adverse mental health. This is consistent with the literature, as previous research showed that insecure residence status can cause postmigration stress [48] and pose a psychological risk [32,49,50]. A study comparing hair cortisol concentration (HCC) of recently fled asylum seekers with and without PTSD found no difference in HCC; however, compared with permanently settled immigrants, recently fled asylum seekers showed higher HCC [51]. This finding has important policy implications, as refugees with insecure residence status suffer from stress, which in turn can negatively affect mental and physical health [52,53]. From a public health perspective, it is therefore critical that legal restrictions on refugees’ access to health care be lifted, regardless of their residence status. The relation of adverse mental health with female sex is established by prior studies [31,54]. As female refugees represent a minority, the lack of gender-specific reception and housing conditions must be addressed. Generally, life in reception centers is more difficult for single women since they lack male protection [55]. In addition, length of stay longer than 36 months since arrival was found to be negatively associated with a higher risk for adverse mental health. A study assessing the prevalence and risk factors for mental distress among refugees in Germany showed that a shorter duration of residence permission was shown to be associated with more severe symptoms of PTSD [56]. Another study focusing on the association between length of stay in asylum centers and mental disorders found that a longer length of stay was associated with an increase in cases of mental disorders [57]. Further data collection and analysis are needed to draw a conclusion. Together these findings suggest that in addition to family separation, sociodemographic and postmigration factors pose major risks for health and well-being among refugees.

There are several limitations to our research. First, we utilized data from the FHS, and selection bias may be an issue because participants in the FHS were self-selected. This means that individuals who were not interested in health issues have decided not to participate in the FHS. In addition, the research questions for this study were developed after the data were collected, so we lack some important information that could be helpful to further understand the relationship between family separation and time pressure. Asking about premigration stressors and reasons for time pressure since their arrival might be beneficial to further understand time pressure as a potential determinant of health among refugees. Another limitation arises in connection with the cross-sectional design. Thus, we were unable to examine the temporal relationship of family separation with time pressure and adverse mental health; longitudinal design studies are recommended. Moreover, there is a need to disentangle the pathways between family separation, time pressure, and mental health through mediation analyses. In addition to that, both outcome variables were dichotomized for regression analyses, resulting in a loss of information [41], particularly for the MCS score; however, the sample mean MCS score was within the range of an optimal cut-off point to screen for adverse mental health such as depressive and anxiety disorders [40] and more than five points below the respective general population mean; thus, it was within the range of five to ten points as a minimally important difference [58]. Since our data only included a convenience sample of 208 refugees from East-Westphalia-Lippe, caution should be taken on the generalizability of the results. It should also be noted that the refugees who were able to come to Germany from Africa and the Middle East might be in relatively better health than refugees who settled in neighboring countries of their home country. Finally, our sample was heterogeneous in terms of origin, causes of flight, and ethnocultural family orientation. Despite its limitations, this study provides new data on subjectively perceived time pressure as a postmigration stressor and on the mental health of refugees from the Middle East and Africa resettled in North Rhine-Westphalia, Germany.

## 5. Conclusions

The study investigated the association of family separation with subjectively perceived time pressure and adverse mental health considering the main relevant confounders in a resettled refugee population in North Rhine-Westphalia, Germany. Participants’ with insecure residence status experienced more time pressure, whereas females, those with an insecure residence status, and those separated from at least one child were at higher risk for adverse mental health. We believe that the present findings help to underscore the importance of family for refugee mental health. Therefore, policies and practices that facilitate family reunification can contribute significantly to the promotion of refugees’ mental health and well-being and thus their integration. Future studies may take a closer look at the temporal relationship of family separation with time pressure and adverse mental health.

## Figures and Tables

**Table 1 ijerph-18-11722-t001:** Characteristics of the study population (*n* = 208).

	*n*	%
**Age**		
Mean (SD)	36.5 (11.2)	
**Sex**		
Male	140	67.3
Female	68	32.7
**Country of origin**		
Syria	87	41.8
Afghanistan	22	10.6
Iraq	58	27.9
Iran	8	3.9
African countries ^a^	14	6.7
Other ^b^	19	9.1
**Residence status**		
Secure	131	63.0
Insecure	68	32.7
*Missing values*	9	4.3
**Length of stay since arrival**		
<12 months	24	11.6
12–24 months	16	7.7
25–36 months	94	45.2
>36 months	71	34.1
*Missing values*	3	1.4
**Separation from spouse or partner**		
Not separated	129	62.0
Separated	57	27.4
*Missing values*	22	10.6
**Separation from child or children**		
Not separated	124	59.6
Separated (from at least one child)	27	13.0
*Missing values*	57	27.4
**Time pressure**		
Time-stressed	81	38.9
Not time-stressed	125	60.1
*Missing values*	2	1.0
**Adverse mental health**		
Higher risk	97	46.6
Lower risk	100	48.1
*Missing values*	11	5.3

*n*, quantity; %, proportion; SD, standard deviation; ^a^ Algeria (14.3%), Eritrea (14.3%), Nigeria (28.6%), Somalia (7.1%), Ghana (14.3%), Morocco (7.1%), Egypt (14.3%); ^b^ Azerbaijan (5.3%), Bangladesh (21.0%), India (5.3%), Lebanon (15.8%), Palestine (10.5%), Russia (10.5%), Tajikistan (5.3%), Turkey (10.5%), stateless (15.8%).

**Table 2 ijerph-18-11722-t002:** Time pressure and adverse mental health groups across independent variables (*n* = 208).

	Time Pressure					Adverse Mental Health				
	Time-Stressed		Not Time-Stressed		*p*	Higher Risk		Lower Risk		*p*
	*n*	%	*n*	%		*n*	%	*n*	%	
**Age**					0.339					0.205
Mean (SD)	35.6 (10.7)		37.1 (11.5)			35.7 (10.5)		37.7 (11.9)		
**Sex**					0.419					0.069
Male	52	37.4	87	62.6		59	44.7	73	55.3	
Female	29	43.3	38	56.7		38	58.5	27	41.5	
**Country of origin**					0.529					0.025
Syria	32	36.8	55	63.2		35	41.2	50	58.8	
Afghanistan	11	50.0	11	50.0		16	80.0	4	20.0	
Iraq	22	38.6	35	61.4		25	45.5	30	54.5	
Iran	5	62.5	3	37.5		6	75.0	2	25.0	
African countries ^a^	6	42.9	8	57.1		6	46.1	7	53.9	
Other ^b^	5	27.8	13	72.2		9	56.3	7	43.7	
**Residence status**					0.161					<0.001
Secure	49	37.4	82	62.6		50	40.0	75	60.0	
Insecure	32	47.8	35	52.2		46	71.9	18	28.1	
**Length of stay since arrival**					0.348					0.279
<12 months	7	29.2	17	70.8		12	52.2	11	47.8	
12–24 months	4	25.0	12	75.0		7	46.7	8	53.3	
25–36 months	39	41.5	55	58.5		50	55.0	41	45.0	
>36 months	31	44.3	39	55.7		26	39.4	40	60.6	
**Separation from spouse or partner**					0.338					0.089
Not separated	47	36.4	82	63.6		54	43.5	70	56.5	
Separated	25	43.9	32	56.1		31	57.4	23	42.6	
**Separation from child or children**					0.957					0.077
Not separated	42	33.9	82	66.1		51	42.5	69	57.5	
Separated (from at least one child)	9	33.3	18	66.7		16	61.5	10	38.5	

*n*, quantity; %, proportion; SD, standard deviation; *p*, *p*-value; ^a^ Algeria, Eritrea, Nigeria, Somalia, Ghana, Morocco, Egypt; ^b^ Azerbaijan, Bangladesh, India, Lebanon, Palestine, Russia, Tajikistan, Turkey, stateless.

**Table 3 ijerph-18-11722-t003:** Results of logistic regression analysis on predictors for being time-stressed (*n* = 129).

Time-Stressed	OR	95% CI		*p*-Value
		Lower	Upper	
**Age**	0.99	0.95	1.03	0.692
**Sex**				
Male	1.00			
Female	1.19	0.48	2.94	0.710
**Country of origin**				
Syria	1.00			
Afghanistan	0.95	0.19	4.73	0.950
Iraq	0.59	0.19	1.82	0.357
Iran	0.98	0.11	8.58	0.983
African countries ^a^	1.14	0.23	5.60	0.874
Other ^b^	0.19	0.03	1.34	0.095
**Residence status**				
Secure	1.00			
Insecure	3.33	1.08	10.28	0.037
**Length of stay since arrival**				
<12 months	1.00			
12–24 months	0.81	0.13	5.08	0.819
25–36 months	1.07	0.29	4.00	0.922
>36 months	1.69	0.46	6.24	0.428
**Separation from spouse or partner**				
Not separated	1.00			
Separated	1.40	0.67	2.91	0.370
**Separation from child or children**				
Not separated	1.00			
Separated (from at least one child)	1.23	0.26	5.82	0.797

OR, odds ratio; CI, confidence interval; ^a^ Algeria, Eritrea, Nigeria, Somalia, Ghana, Morocco, Egypt; ^b^ Azerbaijan, Bangladesh, India, Lebanon, Palestine, Russia, Tajikistan, Turkey, stateless.

**Table 4 ijerph-18-11722-t004:** Results of logistic regression analysis on predictors for higher risk for adverse mental health, *n* = 124.

Adverse Mental Health, Higher Risk	OR	95% CI		*p*-Value
		Lower	Upper	
**Age**	0.99	0.95	1.03	0.471
**Sex**				
Male	1.00			
Female	3.16	1.33	7.51	0.009
**Country of origin**				
Syria	1.00			
Afghanistan	1.64	0.32	8.38	0.552
Iraq	0.82	0.30	2.27	0.704
Iran	2.21	0.20	24.42	0.518
African countries ^a^	0.70	0.13	3.77	0.676
Other ^b^	0.85	0.17	4.28	0.842
**Residence status**				
Secure	1.00			
Insecure	2.96	1.04	8.42	0.042
**Length of stay since arrival**				
<12 months	1.00			
12–24 months	0.40	0.07	2.33	0.308
25–36 months	0.71	0.21	2.45	0.587
>36 months	0.27	0.08	0.99	0.048
**Separation from spouse or partner**				
Not separated	1.00			
Separated	2.10	0.60	7.38	0.247
**Separation from child or children**				
Not separated	1.00			
Separated (from at least one child)	3.53	1.23	10.11	0.019

OR, odds ratio; CI, confidence interval; ^a^ Algeria, Eritrea, Nigeria, Somalia, Ghana, Morocco, Egypt; ^b^ Azerbaijan, Bangladesh, India, Lebanon, Palestine, Russia, Tajikistan, Turkey, stateless.

## Data Availability

The data presented in this study are available from the corresponding author upon reasonable request, due to privacy and ethical reasons.

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
