# Peer review of "Impact of Family Separation on Subjective Time Pressure and Mental Health in Refugees from the Middle East and Africa Resettled in North Rhine-Westphalia, Germany: A Cross-Sectional Study"

_ijerph, 2021, doi:10.3390/ijerph182111722_

Round 1
Reviewer 1 Report
First of all, thank the authors for their research work on social determinants among resettled refugees in Germany, specifically on Impact of family separation on subjective time pressure and mental health. Thus, as the improvements in the paper carried out from previous reviews in which I have participated.
I consider it to be a relevant topic at present, although the paper identifies certain methodological and other limitations to take into account for future research.
As I have commented before, I have identified improvements in the weaknesses that the paper previously presented and I appreciate the references regarding the current situation of the COVID-19 pandemic.
It is for all this that I recommend to the editors the publication of this paper.
Author Response
We thank the reviewer for the reading of our manuscript.
Reviewer 2 Report
No changes needed.
Author Response
We thank you again for your reading of our manuscript.
Reviewer 3 Report
This is a paper addressing family separation, subjective time pressure and mental health in refugees resettled in Germany. Primary data collection with people applying for asylum or having limited stay is a methodological and ethical challenge. The analyses conducted would provide relevant information for policy makes, health care providers as well as researchers. Unfortunately, in the present version of the manuscript, I can’t see the potential to make an important contribution to the literature, so this paper, in its present version, could be seen as suitable for the International Journal of Environmental Research and Public Health audience.
The manuscript has a number of weaknesses, so I would like to highlight most important issues from my perspective:
First of all, since the study is focusing on refugees living in Germany, I miss the research on health status in general as well as on mental component of HRQoL (one of the key outcomes in this study) in different refugee populations in Germany. Indeed, there is a great body of research on refugee’s health status, health care utilization, health care needs being published during last years in Germany only. Here some examples:
Biddle, L., Menold, N., Bentner, M., Nöst, S., Jahn, R., Ziegler, S., & Bozorgmehr, K. (2019). Health monitoring among asylum seekers and refugees: A state-wide, cross-sectional, population-based study in Germany. Emerging Themes in Epidemiology, 16(1), 1-21.
Nesterko, Y., Jäckle, D., Friedrich, M., Holzapfel, L. & Glaesmer, H. (2020). Prevalence of post-traumatic stress disorder, depression and somatisation in recently arrived refugees in Germany: an epidemiological study. Epidemiology and Psychiatric Sciences, 29, e40.
Nesterko, Y., Jäckle, D., Friedrich, M., Holzapfel, L. & Glaesmer, H. (2020). Factors predicting symptoms of somatization, depression, anxiety, post-traumatic stress disorder, self-rated mental and physical health among recently arrived refugees in Germany. Confl Health, 14, 44.
Wetzke, M., Happle, C., Vakilzadeh, A., Ernst, D., Sogkas, G., Schmidt, R. E., … Jablonka, A. (2018). Healthcare Utilization in a Large Cohort of Asylum Seekers Entering Western Europe in 2015. International Journal of Environmental Research and Public Health, 15(10).
In other words, you are focusing on refugees living in Germany, so please refer to existing literature on this host country first.
One more general concern: I think it might be of importance to add some basic statements to your introduction – why is it important to focus on time pressure, family separation and mental health? Why those three factors? Why the combination of those? All the associations the authors refer to in the introduction (and discussion) with social relationships and loneliness aren’t unique to migrant and/or refugee groups at all. You’ll find such associations with native groups as well – which was actually well investigated based on SOEP data using time pressure and SF-12 questionnaire. Please also check on SOEP-based publication in the last years – I believe there are some with special focus on HRQoL in refugees as well.
My major concern is however the operationalization and thus usage of the term family separation. One quick look at numbers in table 1 is enough – the great majority of the sample is young and single. By comparing those being separated from their children with those without children the authors make no statement about impact of possible family separation at all. You need to compare parents (those separated from their children and those not separated) only. Or you compare those in partnership and separated/not separated.. In other words, I’m not convinced by the usage of wording family separation given the way the authors performed the key analyzes. I believe, there is a lot of work on the conceptual framework to do. By the way, there is no information about the model fit indices – so how the readers suppose to handle with the quality of analyzes presented?
And last one – please reconsider you statement in your first sentence. Syrian crisis started a bit earlier than 2015, as well as all of the conflicts in Middle East and some parts of Africa… In 2015, the great majority of refugees crossed the border of EU because of unhuman and unsafe environments in mass camps in Lebanon, Turkey, Jordan, Egypt and Libya as well as restricted asylum policies in these countries starting 2014-2015.
Author Response
Please find below a point-by-point response to all comments (in italics). All page, line and table numbers refer to the revised manuscript, where changes are highlighted.
- The reviewer commented, “This is a paper addressing family separation, subjective time pressure and mental health in refugees resettled in Germany. Primary data collection with people applying for asylum or having limited stay is a methodological and ethical challenge. The analyses conducted would provide relevant information for policy makes, health care providers as well as researchers. Unfortunately, in the present version of the manuscript, I can’t see the potential to make an important contribution to the literature, so this paper, in its present version, could be seen as suitable for the International Journal of Environmental Research and Public Health audience.”
Response: We thank the reviewer for the thoroughly reading of our manuscript and the critical inputs. We have tried to answer all questions to the best of our ability, see below.
- The reviewer commented, “The manuscript has a number of weaknesses, so I would like to highlight most important issues from my perspective: First of all, since the study is focusing on refugees living in Germany, I miss the research on health status in general as well as on mental component of HRQoL (one of the key outcomes in this study) in different refugee populations in Germany. Indeed, there is a great body of research on refugee’s health status, health care utilization, health care needs being published during last years in Germany only. Here some examples: (…). In other words, you are focusing on refugees living in Germany, so please refer to existing literature on this host country first”
Response: Thank you for pointing this out – we now extended the introduction section by adding more information on refugees’ health status, HRQoL, and mental health care needs as well as sources related to the refugee population in Germany (p. 1, ll. 33-40; and p.2, ll. 87-90).
- The reviewer commented, “One more general concern: I think it might be of importance to add some basic statements to your introduction – why is it important to focus on time pressure, family separation and mental health? Why those three factors? Why the combination of those? All the associations the authors refer to in the introduction (and discussion) with social relationships and loneliness aren’t unique to migrant and/or refugee groups at all. You’ll find such associations with native groups as well – which was actually well investigated based on SOEP data using time pressure and SF-12 questionnaire. Please also check on SOEP-based publication in the last years – I believe there are some with special focus on HRQoL in refugees as well.”
Response: We have added more information about why it is important to focus on family separation, time pressure and mental health among refugees resettled in Germany (p. 3, ll. 52-66 and ll. 83-88). We hope it is clearer now.
- The reviewer commented, “My major concern is however the operationalization and thus usage of the term family separation. One quick look at numbers in table 1 is enough – the great majority of the sample is young and single. By comparing those being separated from their children with those without children the authors make no statement about impact of possible family separation at all. You need to compare parents (those separated from their children and those not separated) only. Or you compare those in partnership and separated/not separated. In other words, I’m not convinced by the usage of wording family separation given the way the authors performed the key analyzes. I believe, there is a lot of work on the conceptual framework to do. By the way, there is no information about the model fit indices – so how the readers suppose to handle with the quality of analyzes presented?”
Response: Thank you for this valuable suggestion – we limited the analyses to refugees with a spouse or partner and/or child only (see p. 3, ll. 117-122). Furthermore, we have added the model fit statistics at appropriate places in the text (p. 7, ll. 229-230; p. 8, ll. 240-241).
- The reviewer commented, “And last one – please reconsider you statement in your first sentence. Syrian crisis started a bit earlier than 2015, as well as all of the conflicts in Middle East and some parts of Africa… In 2015, the great majority of refugees crossed the border of EU because of unhuman and unsafe environments in mass camps in Lebanon, Turkey, Jordan, Egypt and Libya as well as restricted asylum policies in these countries starting 2014-2015.”
Response: We have revised the first sentence (p. 1, ll. 28-29).

Reviewer 4 Report
I have major concerns regarding three points. (1) I was unable to comprehend, why the GSOEP item on time pressure was included in the study. It looked a bit like, oh, we are studying effects of time pressure in other studies, let's also include it here. The theoretical basis for including it is rather weak. In German, I would say "Die Variable fällt vom Himmel" (The variable pops up out of the blue) (2) My more serious concern is that is remains totally unclear, why continuous variables were transformed to categorical variables to then perform (if I understood correctly) independent logistic regressions. There may be very good reasons to transform the originally continuous DVs into categorical variables (skewness, may have been one, or, in more general terms, strong deviations from normality), but this must be explained to the reader. Keeping the continuous responses would also have opened the door to a structural equation model with two--simultaneous--dependent variables. (3) My third questions may come a bit as a wise-crack remark, but nevertheless I asked myself, whether and why no pre-migration stress variables were included. Yes, it is a common-place result of migration literature (also in our own work on the topic on Idemudia and Boehnke, 2020) that postmigration stressors are more important, but pre-migration stressors do play a certain role and should not entirely be ignored. All is all, I find this an interesting paper that has a potential, but in it its current state it is more a technical report than a journal article.Author Response
Please find below a point-by-point response to all comments (in italics). All page, line and table numbers refer to the revised manuscript, where changes are highlighted.
- The reviewer commented, “I have major concerns regarding three points. (1) I was unable to comprehend, why the GSOEP item on time pressure was included in the study. It looked a bit like, oh, we are studying effects of time pressure in other studies, let's also include it here. The theoretical basis for including it is rather weak. In German, I would say "Die Variable fällt vom Himmel" (The variable pops up out of the blue)”
Response: Thank you for pointing this out – we now extended the introduction section by adding more information on time pressure as a potentially social determinant of health (p. 2, ll. 48-62). We hope it is clearer now.
- The reviewer commented, “(2) My more serious concern is that is remains totally unclear, why continuous variables were transformed to categorical variables to then perform (if I understood correctly) independent logistic regressions. There may be very good reasons to transform the originally continuous DVs into categorical variables (skewness, may have been one, or, in more general terms, strong deviations from normality), but this must be explained to the reader. Keeping the continuous responses would also have opened the door to a structural equation model with two--simultaneous--dependent variables.”
Response: Previous studies have found that the mental health component of the SF-36 and SF-12 is a useful screening tool for monitoring the prevalence of affective disorders and for targeting treatment and prevention, in which optimal cut-off values vary between a MCS score of 42.0 and 45.6 (see p.3, l. 146). We used the sample mean value = 44.5 as a cut-off point to divide our sample into two approximately equal groups (median was 44.6), and to classify people at lower and higher risk for mental health problems which is consistent with the literature. We then used logistic regression to examine whether or not family separation was associated with a higher risk of adverse mental health.
- The reviewer commented, “(3) My third questions may come a bit as a wise-crack remark, but nevertheless I asked myself, whether and why no pre-migration stress variables were included. Yes, it is a common-place result of migration literature (also in our own work on the topic on Idemudia and Boehnke, 2020) that postmigration stressors are more important, but pre-migration stressors do play a certain role and should not entirely be ignored. All is all, I find this an interesting paper that has a potential, but in it its current state it is more a technical report than a journal article.”
Response: We agree with the reviewer that premigration stress may also play a role. However, the research questions for this study were developed after the data was collected so we lack some important information on premigration stressors. We have now mentioned this aspect in the limitations section (p. 10, l. 331).
We appreciate your careful reading of our manuscript and the critical inputs. We have tried to answer your questions to the best of our ability.

Round 2
Reviewer 3 Report
The authors have properly adressed all the points raised by previous review, so I suggest to accept the manuscriptReviewer 4 Report
Thank you for addressing most of my queries. One can always find additional quibbles, but from my point of view the paper is now in a publishable shape